# ATOMO: Communication-efficient Learning via Atomic Sparsification

**Hongyi Wang**[1]*, **Scott Sievert**[2]*, **Zachary Charles**[2], **Shengchao Liu**[1],
**Stephen Wright**[1], **Dimitris Papailiopoulos**[2]
[1]Department of Computer Sciences,    [2]Department of Electrical and Computer Engineering
University of Wisconsin-Madison

## Abstract

Distributed model training suffers from communication overheads due to frequent gradient updates transmitted between compute nodes. To mitigate these overheads, several studies propose the use of sparsified stochastic gradients. We argue that these are facets of a general sparsification method that can operate on any possible *atomic decomposition*. Notable examples include element-wise, singular value, and Fourier decompositions. We present ATOMO, a general framework for atomic sparsification of stochastic gradients. Given a gradient, an atomic decomposition, and a sparsity budget, ATOMO gives a random unbiased sparsification of the atoms minimizing variance. We show that recent methods such as QSGD and TernGrad are special cases of ATOMO and that sparsifiying the singular value decomposition of neural networks gradients, rather than their coordinates, can lead to significantly faster distributed training.

## 1   Introduction

Several machine learning frameworks such as TensorFlow [1], MXNet [2], and Caffe2[3], come with distributed implementations of popular training algorithms, such as mini-batch SGD. However, the empirical speed-up gains offered by distributed training, often fall short of the optimal linear scaling one would hope for. It is now widely acknowledged that communication overheads are the main source of this speedup saturation phenomenon [4, 5, 6, 7, 8].

Communication bottlenecks are largely attributed to frequent gradient updates transmitted between compute nodes. As the number of parameters in state-of-the-art models scales to hundreds of millions [9, 10], the size of gradients scales proportionally. These bottlenecks become even more pronounced in the context of federated learning [11, 12], where edge devices (*e.g.*, mobile phones, sensors, etc) perform decentralized training, but suffer from low-bandwidth during up-link.

To reduce the cost of of communication during distributed model training, a series of recent studies propose communicating low-precision or sparsified versions of the computed gradients during model updates. Partially initiated by a 1-bit implementation of SGD by Microsoft in [5], a large number of recent studies revisited the idea of low-precision training as a means to reduce communication [13, 14, 15, 16, 17, 18, 19, 17, 20, 21]. Other approaches for low-communication training focus on sparsification of gradients, either by thresholding small entries or by random sampling [6, 22, 23, 24, 25, 26, 27, 28]. Several approaches, including QSGD and TernGrad, implicitly combine quantization and sparsification to maximize performance gains [14, 16, 12, 29, 30], while providing provable guarantees for convergence and performance. We note that quantization methods in the context of gradient based updates have a rich history, dating back to at least as early as the 1970s [31, 32, 33].

---

**Our Contributions** An atomic decomposition represents a vector as a linear combination of simple building blocks in an inner product space. In this work, we show that stochastic gradient sparsification and quantization are facets of a general approach that sparsifies a gradient in any possible atomic decomposition, including its entry-wise or singular value decomposition, its Fourier decomposition, and more. With this in mind, we develop ATOMO, a general framework for atomic sparsification of stochastic gradients. ATOMO sets up and optimally solves a meta-optimization that minimizes the variance of the sparsified gradient, subject to the constraints that it is sparse on the atomic basis, and also is an unbiased estimator of the input.

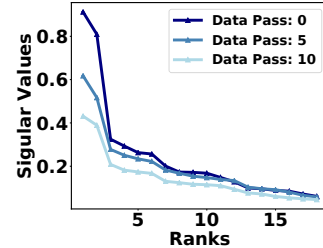

We show that 1-bit QSGD and TernGrad are in fact special cases of ATOMO, and each is optimal (in terms of variance and sparsity), in different parameter regimes. Then, we argue that for some neural network applications, viewing the gradient as a concatenation of matrices (each corresponding to a layer), and applying atomic sparsification to their SVD is meaningful and well-motivated by the fact that these matrices are approximately low-rank (see Fig. 1). We show that ATOMO on the SVD of each layer's gradient, can lead to less variance, and faster training, for the same communication budget as that of QSGD or TernGrad. We present extensive experiments showing that using ATOMO with SVD sparsification can lead to up to $2\times/3\times$ faster training time (including the time to compute the SVD) compared to QSGD/TernGrad. This holds using VGG and ResNet-18 on SVHN and CIFAR-10.

Figure 1: The singular values of a convolutional layer's gradient, for ResNet-18 while training on CIFAR-10. The gradient of a layer can be seen as a matrix, once we vectorize and appropriately stack the conv-filters. For all presented data passes, there is a sharp decay in singular values, with the top 3 standing out.

**Relation to Prior Work** ATOMO is closely related to work on communication-efficient distributed mean estimation in [29] and [30]. These works both note, as we do, that variance (or equivalently the mean squared error) controls important quantities such as convergence, and they seek to find a low-communication vector averaging scheme that minimizes it. Our work differs in two key aspects. First, we derive a closed-form solution to the variance minimization problem for all input gradients. Second, ATOMO applies to any atomic decomposition, which allows us to compare entry-wise against singular value sparsification for matrices. Using this, we derive explicit conditions for which SVD sparsification leads to lower variance for the same sparsity budget.

The idea of viewing gradient sparsification through a meta-optimization lens was also used in [34]. Our work differs in two key ways. First, [34] consider the problem of minimizing the sparsity of a gradient for a fixed variance, while we consider the reverse problem, that is, minimizing the variance subject to a sparsity budget. The second more important difference is that while [34] focuses on entry-wise sparsification, we consider a general problem where we sparsify according to any atomic decomposition. For instance, our approach directly applies to sparsifying the singular values of a matrix, which gives rise to faster training algorithms.

Finally, low-rank factorizations and sketches of the gradients when viewed as matrices were proposed in [35, 36, 37, 38, 12]; arguably most of these methods (with the exception of [12]) aimed to address the high flops required when training low-rank models. Though they did not directly aim to reduce communication, this arises as a useful side effect.

## 2    Problem Setup

In machine learning, we often wish to find a model $w$ minimizing the *empirical risk*

$$f(w) = \frac{1}{n}\sum_{i=1}^{n}\ell(w;x_i) \qquad (1)$$

where $x_i \in \mathbb{R}^d$ is the $i$-th data point. One way to approximately minimize $f(w)$ is by using stochastic gradient methods that operate as follows:

$$w_{k+1} = w_k - \gamma\widehat{g}(w_k)$$

where $w_0$ is some initial model, $\gamma$ is the stepsize, and $\widehat{g}(w)$ is a stochastic gradient of $f(w)$, *i.e.* it is an unbiased estimate of the true gradient $g(w) = \nabla f(w)$. Mini-batch SGD, one of the most common algorithms for distributed training, computes $\widehat{g}$ as an average of $B$ gradients, each evaluated on randomly sampled data from the training set. Mini-batch SGD is easily parallelized in the parameter

server (PS) setup, where a PS stores the global model, and $P$ compute nodes split the effort of computing the $B$ gradients. Once the PS receives these gradients, it applies them to the model, and sends it back to the compute nodes.

To prove convergence bounds for stochastic-gradient based methods, we usually require $\widehat{g}(w)$ to be an unbiased estimator of the full-batch gradient, and to have small variance $\mathbb{E}\|\widehat{g}(w)\|^2$, as this controls the speed of convergence. To see this, suppose $w^*$ is a critical point of $f$, then we have

$$\mathbb{E}[\|w_{k+1} - w^*\|_2^2] = \mathbb{E}[\|w_k - w^*\|_2^2] - \underbrace{\left(2\gamma\langle\nabla f(w_k), w_k - w_*\rangle - \gamma^2\mathbb{E}[\|\widehat{g}(w_k)\|_2^2]\right)}_{\text{progress at step } t}.$$

In particular, the *progress* made by the algorithm at a single step is, in expectation, controlled by the term $\mathbb{E}[\||\widehat{g}(w_k)\||_2^2]$; the smaller it is, the bigger the progress. This is a well-known fact in optimization, and most convergence bounds for stochastic-gradient based methods, including minibatch, involve upper bounds on $\mathbb{E}[\||\widehat{g}(w_k)\||_2^2]$, in a multiplicative form, for both convex and nonconvex setups [39, 40, 41, 42, 42, 43, 44, 45, 46, 47]. Hence, recent results on low-communication variants of SGD design unbiased quantized or sparse gradients, and try to minimize their variance [14, 29, 34].

Since variance is a proxy for speed of convergence, in the context of communication-efficient stochastic gradient methods, one can ask: *What is the smallest possible variance of a stochastic gradient that is represented with $k$ bits?* This can be cast as the following meta-optimization:

$$\min_g \mathbb{E}\|\widehat{g}(w)\|^2$$

$$\text{s.t. } \mathbb{E}[\widehat{g}(w)] = g(w)$$

$$\widehat{g}(w) \text{ can be expressed with } k \text{ bits}$$

Here, the expectation is taken over the randomness of $\widehat{g}$. We are interested in designing a stochastic approximation $\widehat{g}$ that "solves" this optimization. However, it seems difficult to design a formal, tractable version of the last constraint. In the next section, we replace this with a simpler constraint that instead requires that $\widehat{g}(w)$ is sparse with respect to a given atomic decomposition.

## 3  ATOMO: Atomic Decomposition and Sparsification

Let $(V, \langle\cdot, \cdot\rangle)$ be an inner product space over $\mathbb{R}$ and let $\|\cdot\|$ denote the induced norm on $V$. In what follows, you may think of $g$ as a stochastic gradient of the function we wish to optimize. An *atomic decomposition* of $g$ is any decomposition of the form $g = \sum_{a\in\mathcal{A}} \lambda_a a$ for some set of atoms $\mathcal{A} \subseteq V$. Intuitively, $\mathcal{A}$ consists of simple building blocks. We will assume that for all $a \in \mathcal{A}$, $\|a\| = 1$, as this can be achieved by a positive rescaling of the $\lambda_a$.

An example of an atomic decomposition is the entry-wise decomposition $g = \sum_i g_i e_i$ where $\{e_i\}_{i=1}^n$ is the standard basis. More generally, any orthonormal basis of $V$ gives rise to a unique atomic decomposition of $g$. While we focus on finite dimensional vectors, one could use Fourier and wavelet decompositions in this framework. When considering matrices, the singular value decomposition gives an atomic decomposition in the set of rank-1 matrices. More general atomic decompositions have found uses in a variety of situations, including solving linear inverse problems [48].

We are interested in finding an approximation to $g$ with fewer atoms. Our primary motivation is that this reduces communication costs, as we only need to send atoms with non-zero weights. We can use whichever decomposition is most amenable for sparsification. For instance, if $X$ is a low rank matrix, then its singular value decomposition is naturally sparse, so we can save communication costs by sparsifying its singular value decomposition instead of its entries.

Suppose $\mathcal{A} = \{a_i\}_{i=1}^n$ and we have an atomic decomposition $g = \sum_{i=1}^n \lambda_i a_i$. We wish to find an unbiased estimator $\widehat{g}$ of $g$ that is sparse in these atoms, and with small variance. Since $\widehat{g}$ is unbiased, minimizing its variance is equivalent to minimizing $\mathbb{E}[\|\widehat{g}\|^2]$. We use the following estimator:

$$\widehat{g} = \sum_{i=1}^n \frac{\lambda_i t_i}{p_i} a_i \tag{2}$$

where $t_i \sim \text{Bernoulli}(p_i)$, for $0 < p_i \leq 1$. We refer to this sparsification scheme as *atomic sparsification*. Note that the $t_i$'s are independent. Recall that we assumed above that $\|a_i\|^2 = 1$ for all $a_i$. We have the following lemma about $\widehat{g}$.

**Lemma 1.** $\mathbb{E}[\widehat{g}] = g$ *and* $\mathbb{E}[\|\widehat{g}\|^2] = \sum_{i=1}^n \lambda_i^2 p_i^{-1} + \sum_{i\neq j} \lambda_i \lambda_j \langle a_i, a_j\rangle$.

Let $\lambda = [\lambda_1, \ldots, \lambda_n]^T$, $p = [p_1, \ldots, p_n]^T$. In order to ensure that this estimator is sparse, we fix some *sparsity budget* $s$. That is, we require $\sum_i p_i = s$. This is a *sparsity on average* constraint. We wish to minimize $\mathbb{E}[\|\widehat{g}\|^2]$ subject to this constraint. By Lemma 1, this is equivalent to

$$\min_p \sum_{i=1}^n \frac{\lambda_i^2}{p_i} \quad \text{s.t.} \ \forall i, \ 0 < p_i \leq 1, \ \sum_{i=1}^n p_i = s. \tag{3}$$

An equivalent form of this problem was presented in [29] (Section 6.1). The authors considered this problem for entry-wise sparsification and found a closed-form solution for $s \leq \|\lambda\|_1 / \|\lambda\|_\infty$. We give a version of their result but extend it to all $s$. A similar optimization problem was given in [34], which instead minimizes sparsity subject to a variance constraint.

---

**Algorithm 1:** ATOMO probabilities

---

**Input** : $\lambda \in \mathbb{R}^n$ with $|\lambda_1| \geq \ldots |\lambda_n|$; sparsity budget $s$ such that $0 < s \leq n$.
**Output:** $p \in \mathbb{R}^n$ solving (3).
$i = 1$;
**while** $i \leq n$ **do**
    **if** $|\lambda_i|s \leq \sum_{j=i}^n |\lambda_i|$ **then**
        **for** $k = i, \ldots, n$ **do**
            $p_k = |\lambda_k|s \left( \sum_{j=i}^n |\lambda_i| \right)^{-1}$;
        **end**
        $i = n + 1$;
    **else**
        $p_i = 1, s = s - 1$;
        $i = i + 1$;
    **end**
**end**

---

We will show that Algorithm 1 produces $p \in \mathbb{R}^n$ solving (3). While we show in Appendix B that this can be derived via the KKT conditions, we focus on an alternative method relaxes (3) to better understand its structure. This approach also analyzes the variance achieved by solving (3) more directly.

Note that (3) has non-empty feasible set only for $0 < s \leq n$. Define $f(p) := \sum_{i=1}^n \lambda_i^2 / p_i$. To understand how to solve (3), we first consider the following relaxation:

$$\min_p \sum_{i=1}^n \frac{\lambda_i^2}{p_i} \quad \text{s.t.} \ \forall i, \ 0 < p_i, \ \sum_{i=1}^n p_i = s. \tag{4}$$

We have the following lemma about the solutions to (4), first shown in [29].

**Lemma 2** ([29]). *Any feasible vector $p$ to* (4) *satisfies* $f(p) \geq \dfrac{1}{s}\|\lambda\|_1^2$. *This is achieved iff* $p_i = \dfrac{|\lambda_i|s}{\|\lambda\|_1}$.

Lemma 2 implies that if we ignore the constraint that $p_i \leq 1$, then the optimal $p$ is achieved by setting $p_i = |\lambda_i|s/\|\lambda\|_1$. If the quantity in the right-hand side is greater than 1, this does not give us an actual probability. This leads to the following definition.

**Definition 1.** *An atomic decomposition $g = \sum_{i=1}^n \lambda_i a_i$ is $s$-unbalanced at entry $i$ if $|\lambda_i|s > \|\lambda\|_1$.*

We say that $g$ is $s$-balanced otherwise. Clearly, an atomic decomposition is $s$-balanced iff $s \leq \|\lambda\|_1 / \|\lambda\|_\infty$. Lemma 2 gives us the optimal way to sparsify $s$-balanced vectors, since the optimal $p$ for (4) is feasible for (3). If $g$ is $s$-unbalanced at entry $j$, we cannot assign this $p_j$ as it is larger than 1. In the following lemma, we show that in $p_j = 1$ is optimal in this setting.

**Lemma 3.** *Suppose that $g$ is $s$-unbalanced at entry $j$ and that $q$ is feasible in* (3). *Then $\exists p$ that is feasible such that $f(p) \leq f(q)$ and $p_j = 1$.*

Let $\phi(g) = \sum_{i \neq j} \lambda_i \lambda_j \langle a_i, a_j \rangle$. Lemmas 2 and 3 imply the following theorem about solutions to (3).

**Theorem 4.** *If $g$ is $s$-balanced, then $\mathbb{E}[\|\widehat{g}\|^2] \geq s^{-1}\|\lambda\|_1^2 + \phi(g)$ with equality if and only if $p_i = |\lambda_i|s/\|\lambda\|_1$. If $g$ is $s$-unbalanced, then $\mathbb{E}[\|\widehat{g}\|^2] > s^{-1}\|\lambda\|_1^2 + \phi(g)$ and is minimized by $p$ with $p_j = 1$ where $j = \text{argmax}_{i=1,\ldots,n} |\lambda_i|$.*

Due to the sorting requirement in the input, Algorithm 1 requires $O(n \log n)$ operations. In Appendix B we describe a variant that uses only $O(sn)$ operations. Thus, we can solve (3) in $O(\min\{n, s\} \log(n))$ operations.

## 4 Relation to QSGD and TernGrad

In this section, we will discuss how ATOMO is related to two recent quantization schemes, 1-bit QSGD [14] and TernGrad [16]. We will show that in certain cases, these schemes are versions of the ATOMO for a specific sparsity budget $s$. Both schemes use the entry-wise atomic decomposition.

QSGD takes as input $g \in \mathbb{R}^n$ and $b \geq 1$. This $b$ governs the number of quantization buckets. When $b = 1$, QSGD produces a random vector $Q(g)$ defined by

$$Q(g)_i = \|g\|_2 \operatorname{sign}(g_i)\zeta_i.$$

Here, the $\zeta_i \sim \text{Bernoulli}(|g_i|/\|g\|_2)$ are independent random variables. One can show this is equivalent to (2) with $p_i = |g_i|/\|g\|_2$ and sparsity budget $s = \|g\|_1/\|g\|_2$. Note that by definition, any $g$ is $s$-balanced for this $s$. Therefore, Theorem 4 implies that the optimal way to assign $p_i$ with this given $s$ is $p_i = |g_i|/\|g\|_2$, which agrees with 1-bit QSGD.

TernGrad takes $g \in \mathbb{R}^n$ and produces a sparsified version $T(g)$ given by

$$T(g)_i = \|g\|_\infty \operatorname{sign}(g_i)\zeta_i$$

where $\zeta_i \sim \text{Bernoulli}(|g_i|/\|g\|_\infty)$. This is equivalent to (2) with $p_i = |g_i|/\|g\|_\infty$ and sparsity budget $s = \|g\|_1/\|g\|_\infty$. Once again, any $g$ is $s$-balanced for this $s$ by definition. Therefore, Theorem 4 implies that the optimal assignment of the $p_i$ for this $s$ is $p_i = |g_i|/\|g\|_\infty$, which agrees with TernGrad.

We can generalize both of these with the following quantization method. Fix $q \in (0, \infty]$. Given $g \in \mathbb{R}^n$, we define the $\ell_q$-quantization of $g$, denoted $L_q(g)$, by

$$L_q(v)_i = \|g\|_q \operatorname{sign}(g_i)\zeta_i$$

where $\zeta_i \sim \text{Bernoulli}(|g_i|/\|g\|_q)$. By the reasoning above, we derive the following theorem.

**Theorem 5.** *$\ell_q$-quantization performs atomic sparsification in the standard basis with $p_i = |g_i|/\|g\|_q$. This solves (3) for $s = \|g\|_1/\|g\|_q$ and satisfies $\mathbb{E}[\|L_q(g)\|_2^2] = \|g\|_1\|g\|_q$.*

In particular, for $q = 2$ we get 1-bit QSGD while for $q = \infty$, we get TernGrad.

## 5 Spectral-ATOMO: Sparsifying the Singular Value Decomposition

For a rank $r$ matrix $X$, denote its singular value decomposition (SVD) by $X = \sum_{i=1}^r \sigma_i u_i v_i^T$. Let $\sigma = [\sigma_1, \ldots, \sigma_r]^T$. We define the $\ell_{p,q}$ norm of a matrix $X$ by $\|X\|_{p,q} = (\sum_{j=1}^m (\sum_{i=1}^n |X_{i,j}|^p)^{q/p})^{1/q}$. When $p = q = \infty$, we define this to be $\|X\|_{\max}$ where $\|X\|_{\max} := \max_{i,j} |X_{i,j}|$.

Let $V$ be the space of real $n \times m$ matrices. Given $X \in V$, there are two standard atomic decompositions of $X$. The first is the entry-wise decomposition $X = \sum_{i,j} X_{i,j} e_i e_j^T$. The second is its SVD $X = \sum_{i=1}^r \sigma_i u_i v_i^T$. If $r$ is small, it may be more efficient to communicate the $r(n+m)$ entries of the SVD, rather than the $nm$ entries of the matrix. Let $\widehat{X}$ and $\widehat{X}_\sigma$ denote the random variables in (2) corresponding to the entry-wise decomposition and singular value decomposition of $X$, respectively. We wish to compare these two sparsifications.

Table 1: Communication cost and variance of ATOMO for matrices.

| Decomposition | Comm. | Var. |
|---|---|---|
| Entry-wise | $s$ | $\dfrac{1}{s}\|X\|_{1,1}^2$ |
| SVD | $s(n+m)$ | $\dfrac{1}{s}\|X\|_*^2$ |

In Table 1, we compare the communication cost and variance of these two methods. The communication cost is the expected number of non-zero elements (real numbers) that need to be communicated. For $\widehat{X}$, a sparsity budget of $s$ corresponds to $s$ non-zero entries we need to communicate. For $\widehat{X}_\sigma$, a sparsity budget of $s$ gives a communication cost of $s(n+m)$ due to the singular vectors. We compare the optimal variance from Theorem 4.

To compare the variance of these two methods under the same communication cost, we want $X$ to be $s$-balanced in its entry-wise decomposition. This holds iff $s \leq \|X\|_{1,1}/\|X\|_{\max}$. By Theorem 4, this gives $\mathbb{E}[\|\widehat{X}\|_F^2] = s^{-1}\|X\|_{1,1}^2$. To achieve the same communication cost with $\widehat{X}_\sigma$, we take a sparsity budget of $s' = s/(n+m)$. The SVD of $X$ is $s'$-balanced iff $s' \leq \|X\|_*/\|X\|_2$. By Theorem 4, $\mathbb{E}[\|\widehat{X}_\sigma\|_F^2] = (n+m)s^{-1}\|X\|_*^2$. This leads to the following theorem.

**Theorem 6.** *Suppose $X \in \mathbb{R}^{n \times m}$ and*

$$s \leq \min\left\{\frac{\|X\|_{1,1}}{\|X\|_{\max}}, (n+m)\frac{\|X\|_*}{\|X\|_2}\right\}.$$

Then $\widehat{X}_\sigma$ with sparsity $s' = s/(n+m)$ incurs the same communication cost as $\widehat{X}$ with sparsity $s$, and $\mathbb{E}[\|\widehat{X}_\sigma\|^2] \leq \mathbb{E}[\|\widehat{X}\|^2]$ if and only if $(n+m)\|X\|_*^2 \leq \|X\|_{1,1}^2$.

To better understand this condition, we will make use of the following well-known fact.

**Lemma 7.** *For any $n \times m$ matrix $X$ over $\mathbb{R}$, $\frac{1}{\sqrt{nm}}\|X\|_{1,1} \leq \|X\|_* \leq \|X\|_{1,1}$.*

For expository purposes, we give a proof of this Appendix C and show that these bounds are the best possible. As a result, if the first inequality is tight, then $\mathbb{E}[\|\widehat{X}_\sigma\|^2] \leq \mathbb{E}[\|\widehat{X}\|^2]$, while if the second is tight then $\mathbb{E}[\|\widehat{X}_\sigma\|^2] \geq \mathbb{E}[\|\widehat{X}\|^2]$. As we show in the next section, using singular value sparsification can translate in to significantly reduced distributed training time.

## 6 Experiments

We present an empirical study of Spectral-ATOMO and compare it to the recently proposed QSGD [14], and TernGrad [16], on a different neural network models and data sets, under real distributed environments. Our main findings are as follows:

- We observe that spectral-ATOMO provides a useful alternative to entry-wise sparsification methods, it reduces communication compared to vanilla mini-batch SGD, and can reduce training time compared to QSGD and TernGrad by up to a factor of $2\times$ and $3\times$ respectively. For instance, on VGG11-BN trained on CIFAR-10, spectral-ATOMO with sparsity budget 3 achieves $3.96\times$ speedup over vanilla SGD, while 4-bit QSGD achieves $1.68\times$ on a cluster of 16, g2.2xlarge instances. Both ATOMO and QSGD greatly outperform TernGrad as well.

- We observe that spectral-ATOMO in distributed settings leads to models with negligible accuracy loss when combined with parameter tuning.

**Implementation and setup** We compare spectral-ATOMO[2] with different sparsity budgets to $b$-bit QSGD across a distributed cluster with a parameter server (PS), implemented in mpi4py [49] and PyTorch [50] and deployed on multiple types of instances in Amazon EC2 (*e.g.*m5.4xlarge, m5.2xlarge, and g2.2xlarge), both PS and compute nodes are of the same type of instance. The PS implementation is standard, with a few important modifications. At the most basic level, it receives gradients from the compute nodes and broadcasts the updated model once a batch has been received.

In our experiments, we use data augmentation (random crops, and flips), and tuned the step-size for every different setup as shown in Table 5 in Appendix D. Momentum and regularization terms are switched off to make the hyperparamter search tractable and the results more legible. Tuning the step sizes for this distributed network for three different datasets and eight different coding schemes can be computationally intensive. As such, we only used small networks so that multiple networks could fit into GPU memory. To emulate the effect of larger networks, we use synchronous message communication, instead of asynchronous.

Each compute node evaluates gradients sampled from its partition of data. Gradients are then sparsified through QSGD or spectral-ATOMO, and then are sent back to the PS. Note that spectral-ATOMO transmits the weighted singular vectors sampled from the true gradient of a layer. The PS then combines these, and updates the model with the average gradient. Our entire experimental pipeline is implemented in PyTorch [50] with mpi4py [49], and deployed on either g2.2xlarge, m5.2xlarge and m5.4xlarge instances in Amazon AWS EC2. We conducted our experiments on various models, datasets, learning tasks, and neural network models as detailed in Table 2.

| Dataset | CIFAR-10 | CIFAR-100 | SVHN |
|---|---|---|---|
| # Data points | 60,000 | 60,000 | 600,000 |
| Model | ResNet-18 / VGG-11-BN | ResNet-18 | ResNet-18 |
| # Classes | 10 | 100 | 10 |
| # Parameters | 11,173k / 9,756k | 11,173k | 11,173k |

Table 2: The datasets used and their associated learning models and hyper-parameters.

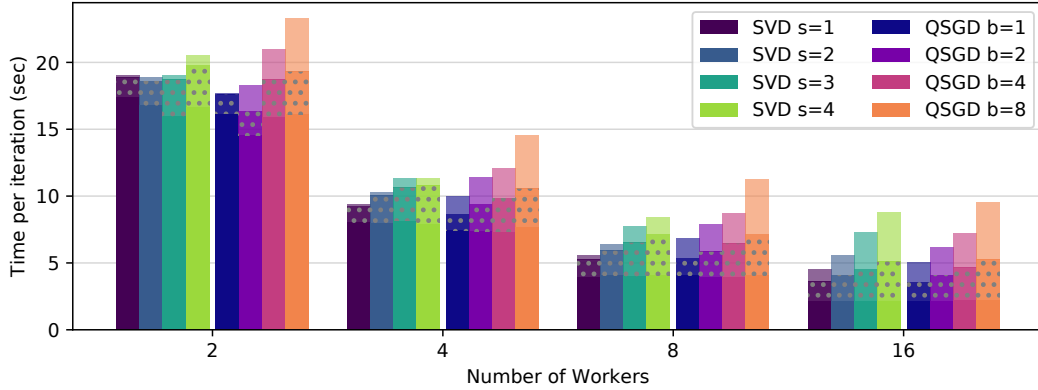

Figure 2: The timing of the gradient coding methods (QSGD and spectral-ATOMO) for different quantization levels, $b$ bits and $s$ sparsity budget respectively for each worker when using a ResNet-34 model on CIFAR10. For brevity, we use SVD to denote spectral-ATOMO. The bars represent the total iteration time and are divided into computation time (bottom, solid), encoding time (middle, dotted) and communication time (top, faded).

**Scalability** We study the scalability of these sparsification methods on clusters of different sizes. We used clusters with one PS and $n = 2, 4, 8, 16$ compute nodes. We ran ResNet-34 on CIFAR-10 using mini-batch SGD with batch size $512$ split among compute nodes. The experiment was run on m5.4xlarge instances of AWS EC2 and the results are shown in Figure 2.

While increasing the size of the cluster, decreases the computational cost per worker, it causes the communication overhead to grow. We denote as computational cost, the time cost required by each worker for gradient computations, while the communication overhead is represented by the amount time the PS waits to receive the gradients by the slowest worker. This increase in communication cost is non-negligible, even for moderately-sized networks with sparsified gradients. We observed a trade-off in both sparsification approaches between the information retained in the messages after sparsification and the communication overhead.

**End-to-end convergence performance** We evaluate the end-to-end convergence performance on different datasets and neural networks, training with spectral-ATOMO(with sparsity budget $s = 1, 2, 3, 4$), QSGD (with $n = 1, 2, 4, 8$ bits), and ordinary mini-batch SGD. The datasets and models are summarized in Table 2. We use ResNet-18 [9] and VGG11-BN [51] for CIFAR-10 [52] and SVHN [53]. Again, for each of these methods we tune the step size. The experiments were run on a cluster of 16 compute nodes instantiated on g2.2xlarge instances.

The gradients of convolutional layers are 4 dimensional tensors with shape of $[x, y, k, k]$ where $x, y$ are two spatial dimensions and $k$ is the size of the convolutional kernel. However, matrices are required to compute the SVD for spectral-ATOMO, and we choose to reshape each layer into a matrix of size $[xy/2, 2k^2]$. This provides more flexibility on the sparsity budget for the SVD sparsification. For QSGD, we use the bucketing and Elias recursive coding methods proposed in [14], with bucket size equal to the number of parameters in each layer of the neural network.

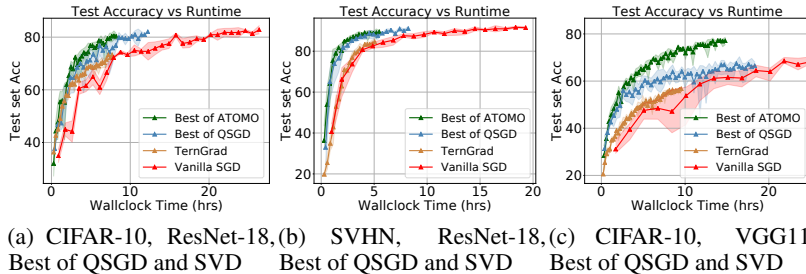

(a) CIFAR-10, ResNet-18, Best of QSGD and SVD
(b) SVHN, ResNet-18, Best of QSGD and SVD
(c) CIFAR-10, VGG11, Best of QSGD and SVD

Figure 3: Convergence rates for the best performance of QSGD and spectral-ATOMO, alongside TernGrad and vanilla SGD. (a) uses ResNet-18 on CIFAR-10, (b) uses ResNet-18 on SVHN, and (c) uses VGG-11-BN on CIFAR-10. For brevity, we use SVD to denote spectral-ATOMO.

Figure 3 shows how the testing accuracy varies with wall clock time. Tables 3 and 4 give a detailed account of speedups of singular value sparsification compared to QSGD. In these tables, each method is run until a specified accuracy.

| Test accuracy | SVD s=1 | SVD s=2 | QSGD b=1 | QSGD b=2 | TernGrad |
|---|---|---|---|---|---|
| 60% | 3.06x | 3.51x | 2.19x | 2.31x | 1.45x |
| 63% | 3.67x | 3.6x | 1.88x | 2.22x | 1.65x |
| 65% | 3.01x | 3.6x | 1.46x | 2.21x | 2.19x |
| 68% | 2.36x | 2.78x | 1.15x | 2.01x | 1.77x |

| Test accuracy | SVD s=3 | SVD s=4 | QSGD b=4 | QSGD b=8 | TernGrad |
|---|---|---|---|---|---|
| 65% | 2.63x | 1.84x | 2.62x | 1.79x | 2.19x |
| 71% | 2.81x | 2.04x | 1.81x | 2.62x | 1.22x |
| 75% | 2.01x | 1.79x | 1.41x | 1.78x | 1.18x |
| 78% | 1.81x | 1.8x | 1.67x | 1.73x | N/A |

Table 3: Speedups of spectral-ATOMO with sparsity budget $s$, $b$-bit QSGD, and TernGrad using ResNet-18 on CIFAR10 over vanilla SGD. N/A stands for the method fails to reach a certain Test accuracy in fixed iterations.

| Test accuracy | SVD s=3 | SVD s=4 | QSGD b=4 | QSGD b=8 | TernGrad |
|---|---|---|---|---|---|
| 75% | 3.55x | 2.75x | 3.22x | 2.36x | 1.33x |
| 78% | 2.84x | 2.75x | 2.68x | 1.89x | 1.23x |
| 82% | 2.95x | 2.28x | 2.23x | 2.35x | 1.18x |
| 84% | 3.11x | 2.39x | 2.34x | 2.35x | 1.34x |

| Test accuracy | SVD s=3 | SVD s=4 | QSGD b=4 | QSGD b=8 | TernGrad |
|---|---|---|---|---|---|
| 85% | 3.15x | 2.43x | 2.67x | 2.35x | 1.21x |
| 86% | 2.58x | 2.19x | 2.29x | 2.1x | N/A |
| 88% | 2.58x | 2.19x | 1.69x | 2.09x | N/A |
| 89% | 2.72x | 2.27x | 2.11x | 2.14x | N/A |

Table 4: Speedups of spectral-ATOMO with sparsity budget $s$ and $b$-bit QSGD, and TernGrad using ResNet-18 on SVNH over vanilla SGD. N/A stands for the method fails to reach a certain Test accuracy in fixed iterations.

We observe that QSGD and ATOMO speed up model training significantly and achieve similar accuracy to vanilla mini-batch SGD. We also observe that the best performance is not achieve with the most sparsified, or quantized method, but the optimal method lies somewhere in the middle where enough information is preserved during the sparsification. For instance, 8-bit QSGD converges faster than 4-bit QSGD, and spectral-ATOMO with sparsity budget 3, or 4 seems to be the fastest. Higher sparsity can lead to a faster running time, but extreme sparsification can adversely affect convergence. For example, for a fixed number of iterations, 1-bit QSGD has the smallest time cost, but may converge much more slowly to an accurate model.

## 7 Conclusion

In this paper, we present and analyze ATOMO, a general sparsification method for distributed stochastic gradient based methods. ATOMO applies to any atomic decomposition, including the entry-wise and the SVD of a matrix. ATOMO generalizes 1-bit QSGD and TernGrad, and provably minimizes the variance of the sparsified gradient subject to a sparsity constraint on the atomic decomposition. We focus on the use ATOMO for sparsifying matrices, especially the gradients in neural network training. We show that applying ATOMO to the singular values of these matrices can lead to faster training than both vanilla SGD or QSGD, for the same communication budget. We present extensive experiments showing that ATOMO can lead to up to a $2\times$ speed-up in training time over QSGD and up to $3\times$ speed-up in training time over TernGrad.

In the future, we plan to explore the use of ATOMO with Fourier decompositions, due to its utility and prevalence in signal processing. More generally, we wish to investigate which atomic sets lead to reduced communication costs. We also plan to examine how we can sparsify and compress gradients in a joint fashion to further reduce communication costs. Finally, when sparsifying the SVD of a matrix, we only sparsify the singular values. We also note that it would be interesting to explore jointly sparsification of the SVD and and its singular vectors, which we leave for future work.

## Footnotes

[2]code available at: `https://github.com/hwang595/ATOMO`

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
