[Supplementary Material]

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

# A  Proof of results

## A.1  Proof of Lemma 2

*Proof.* Suppose we have some $p$ satisfying the conditions in (4). We define two auxiliary vectors $\alpha, \beta \in \mathbb{R}^n$ by

$$\alpha_i = \frac{|\lambda_i|}{\sqrt{p_i}}.$$
$$\beta_i = \sqrt{p_i}.$$

Then note that using the fact that $\sum_i p_i = s$, we have

$$f(p) = \sum_{i=1}^n \frac{\lambda_i^2}{p_i} = \left(\sum_{i=1}^n \frac{\lambda_i^2}{p_i}\right)\left(\frac{1}{s}\sum_{i=1}^n p_i\right) = \frac{1}{s}\left(\sum_{i=1}^n \frac{\lambda_i^2}{p_i}\right)\left(\sum_{i=1}^n p_i\right) = \frac{1}{s}\|\alpha\|_2^2\|\beta\|_2^2.$$

By the Cauchy-Schwarz inequality, this implies

$$f(p) = \frac{1}{s}\|\alpha\|_2^2\|\beta\|_2^2 \geq \frac{1}{s}|\langle \alpha, \beta\rangle|^2 = \frac{1}{s}\left(\sum_{i=1}^n |\lambda_i|\right)^2 = \frac{1}{s}\|\lambda\|_1^2. \tag{5}$$

This proves the first part of Lemma 2. In order to have $f(p) = \frac{1}{s}\|\lambda\|_1^2$, (5) implies that we need

$$|\langle \alpha, \beta\rangle| = \|\alpha\|_2\|\beta\|_2.$$

By the Cauchy-Schwarz inequality, this occurs iff $\alpha$ and $\beta$ are linearly dependent. Therefore, $c\alpha = \beta$ for some constant $c$. Solving, this implies $p_i = c|\lambda_i|$. Since $\sum_{i=1}^n p_i = s$, we have

$$c\|\lambda\|_1 = \sum_{i=1}^n c|\lambda_i| = \sum_{i=1}^n p_i = s.$$

Therefore, $c = \|\lambda\|_1/s$, which implies the second part of the theorem. $\qquad\square$

## A.2  Proof of Lemma 3

Fix $q$ that is feasible in (3). To prove Lemma 3 we will require a lemma. Given the atomic decomposition $g = \sum_{i=1}^n \lambda_i a_i$, we say that $\lambda$ is $s$-unbalanced at $i$ if $|\lambda_i|s > \|\lambda\|_1$, which is equivalent to $g$ being unbalanced in this atomic decomposition at $i$. For notational simplicity, we will assume that $\lambda$ is $s$-unbalanced at $i = 1$. Let $A \subseteq \{2, \ldots, n\}$. We define the following notation:

$$s_A = \sum_{i \in A} q_i.$$

$$f_A(q) = \sum_{i \in A} \frac{\lambda_i^2}{q_i}.$$

$$(\lambda_A)_i = \begin{cases} \lambda_i, & \text{for } i \in A, \\ 0, & \text{for } i \notin A. \end{cases}$$

Note that under this notation, Lemma 2 implies that for all $p > 0$,

$$f_A(p) \geq \frac{1}{s_A}\|\lambda_A\|_1^2. \tag{6}$$

**Lemma 8.** *Suppose that $q$ is feasible and that there is some set $A \subseteq \{2, \ldots, n\}$ such that*

  1. *$\lambda_A$ is $(s_A + q_1 - 1)$-balanced.*

  2. *$|\lambda_1|(s_A + q_1 - 1) > \|\lambda_A\|_1$.*

*Then there is a vector $p$ that is feasible satisfying $f(p) \leq f(q)$ and $p_1 = 1$.*

*Proof.* Suppose that such a set $A$ exists. Let $B = \{2, \ldots, n\} \backslash A$. Note that we have

$$f(q) = \sum_{i=1}^{n} \frac{\lambda_i^2}{q_i} = \frac{\lambda_1^2}{q_1} + f_A(q) + f_B(q).$$

By (6), this implies

$$f(q) \geq \frac{\lambda_1^2}{q_1} + \frac{1}{s_A} \|\lambda_A\|_1^2 + f_B(q). \tag{7}$$

Define $p$ as follows.

$$p_i = \begin{cases} 1, & \text{for } i = 1, \\ \dfrac{|\lambda_i|(s_A + q_1 - 1)}{\|\lambda_A\|_1}, & \text{for } i \in A, \\ q_i, & \text{for } i \notin A. \end{cases}$$

Note that by Assumption 1 and Lemma 2, we have

$$f_A(p) = \frac{1}{s_A + q_1 - 1} \|\lambda_A\|_1^2.$$

Since $p_i = q_i$ for $i \in B$, we have $f_B(p) = f_B(q)$. Therefore,

$$f(p) = \lambda_1^2 + \frac{1}{s_A + q_1 - 1} \|\lambda_A\|_1^2 + f_B(q). \tag{8}$$

Combining (7) and (8), we have

$$f(q) - f(p) = \lambda_1^2 \left( \frac{1}{q_1} - 1 \right) + \|\lambda_A\|_1^2 \left( \frac{1}{s_A} - \frac{1}{s_A + q_1 - 1} \right)$$

$$= \lambda_1^2 \left( \frac{1 - q_1}{q_1} \right) + \|\lambda_A\|_1^2 \left( \frac{q_1 - 1}{s_A(s_A + q_1 - 1)} \right).$$

Combining this with Assumption 2, we have

$$f(q) - f(p) \geq \frac{\|\lambda_A\|_1^2}{(s_A + q_1 - 1)^2} \left( \frac{1 - q_1}{q_1} \right) + \|\lambda_A\|_1^2 \left( \frac{q_1 - 1}{s_A(r_A + q_1 - 1)} \right). \tag{9}$$

To show that the RHS of (9) is at most 0, it suffices to show

$$s_A \geq q_1(s_A + q_1 - 1). \tag{10}$$

However, note that since $0 < q_1 < 1$, the RHS of (10) satisfies

$$q_1(s_A + q_1 - 1) = s_A q_1 - q_1(1 - q_1) \leq s_A q_1 \leq s_A.$$

Therefore, (10) holds, completing the proof. $\square$

We can now prove Lemma 3. In the following, we will refer to Conditions 1 and 2, relative to some set $A$, as the conditions required by Lemma 8.

*Proof.* We first show this in the case that $n = 2$. Here we have the atomic decomposition

$$g = \lambda_1 a_1 + \lambda_2 a_2.$$

The condition that $\lambda$ is $s$-unbalanced at $i = 1$ implies

$$|\lambda_1|(s - 1) > |\lambda_2|.$$

In particular, this implies $s > 1$. For $A = \{2\}$, Condition 1 is equivalent to

$$|\lambda_2|(s_A + q_1 - 2) \leq 0.$$

Note that $s_A = q_2$ and that $q_1 + q_2 - 2 = s - 2$ by assumption. Since $q_i \leq 1$, we know that $s - 2 \leq 0$ and so Condition 1 holds. Similarly, Condition 2 becomes

$$|\lambda_1|(s - 1) > |\lambda_2|$$

which holds by assumption. Therefore, Lemma 3 holds for $n = 2$.

Now suppose that $n > 2$, $q$ is some feasible probability vector, and that $\lambda$ is $s$-unbalanced at index 1. We wish to find an $A$ satisfying Conditions 1 and 2. Consider $B = \{2, \ldots, n\}$. Note that for such $B$, $s_B + q_1 - 1 = s - 1$. By our unbalanced assumption, we know that Condition 2 holds for $B = \{2, \ldots, n\}$. If $\lambda_B$ is $(s-1)$-balanced, then Lemma 8 implies that we are done.

Assume that $\lambda_B$ is not $(s-1)$-balanced. After relabeling, we can assume it is unbalanced at $i = 2$. Let $C = \{3, \ldots, n\}$. Therefore,

$$|\lambda_2|(s-2) > \|\lambda_C\|_1. \tag{11}$$

Combining this with the $s$-unbalanced assumption at $i = 1$, we find

$$
\begin{aligned}
|\lambda_1| &> \frac{\|\lambda_B\|_1}{s-1} \\
&= \frac{|\lambda_2|}{s-1} + \frac{\|\lambda_C\|_1}{s-1} \\
&> \frac{\|\lambda_C\|_1}{(s-1)(s-2)} + \frac{\|\lambda_C\|_1}{s-1} \\
&= \frac{\|\lambda_C\|_1}{s-2}.
\end{aligned}
$$

Therefore,

$$|\lambda_1|(s - q_2 - 1) \geq |\lambda_1|(s-2) > \|\lambda_C\|_1. \tag{12}$$

Let $D = \{1, 3, 4, \ldots, n\} = \{1, \ldots, n\}\backslash\{2\}$. Then note that (12) implies that $\lambda_D$ is $(s-q_2)$-unbalanced at $i = 1$. Inductively applying this theorem, this means that we can find a vector $p' \in \mathbb{R}^{|D|}$ such that $p'_1 = 1$ and $f_D(p') \leq f_D(q)$. Moreover, $s_D(p') = s - q_2$. Therefore, if we let $p$ be the vector that equals $p'$ on $D$ and with $p_2 = q_2$, we have

$$f(p_2) = f_C(p') + \frac{\lambda_2^2}{q_2} \leq f_D(q) + \frac{\lambda_2^2}{q_2} = f(q).$$

This proves the desired result. $\qquad\square$

## B   Analysis of ATOMO via the KKT Condtions

In this section we show how to derive Algorithm 1 using the KKT conditions. Recall that we wish to solve the following optimization problem:

$$\min_p \ f(p) := \sum_{i=1}^{n} \frac{\lambda_i^2}{p_i} \quad \text{subject to } \forall i, \ 0 < p_i \leq 1, \ \sum_{i=1}^{n} p_i = s. \tag{13}$$

We first note a few immediate consequences.

1. If $s > n$ then the problem is infeasible. Note that when $s \geq n$, the optimal thing to do is to set all $p_i = 1$, in which case no sparsification takes place.

2. If $\lambda_i = 0$, then $p_i = 0$. This follows from the fact that this $p_i$ does not change the value of $f(p)$, and the objective could be decreased by allocating more to the $p_j$ associated to non-zero $\lambda_j$. Therefore we can assume that all $\lambda_i \neq 0$.

3. If $|\lambda_i| \geq |\lambda_j| > 0$, then we can assume $p_i \geq p_j$. Otherwise, suppose $p_j > p_i$ but $|\lambda_i| \geq |\lambda_j|$. Let $p'$ denote the vector with $p_i, p_j$ switched. We then have

$$
\begin{aligned}
f(p) - f(p') &= \frac{\lambda_i^2 - \lambda_j^2}{p_i} + \frac{\lambda_j^2 - \lambda_i^2}{p_j} \\
&= \lambda_i^2\left(\frac{1}{p_i} - \frac{1}{p_j}\right) - \lambda_j^2\left(\frac{1}{p_i} - \frac{1}{p_j}\right) \\
&\geq 0.
\end{aligned}
$$

We therefore assume $0 < s \leq n$ and $|\lambda_1| \geq |\lambda_2| \geq \ldots \geq |\lambda_n| > 0$. As above we define $\lambda := [\lambda_1, \ldots, \lambda_n]^T$. While the formulation of (13) does not allow direct application of the KKT conditions, since we have a strict inequality of $0 < p_i$, this is fixed with the following lemma.

**Lemma 9.** *The minimum of* (13) *is achieved by some $p^*$ satisfying*

$$p_i^* \geq \frac{s\lambda_i^2}{n\|\lambda\|_2^2}.$$

*Proof.* Define $\overline{p}$ by $\overline{p}_i = s/n$. This vector is clearly feasible in (13). Let $p$ be any feasible vector. If $f(p) \leq f(q)$ then for any $i \in [n]$ we have

$$\frac{\lambda_i^2}{p_i} \leq f(p) \leq f(\overline{p}).$$

Therefore, $p_i \geq \lambda_i^2/f(\overline{p})$. A straightforward computations shows that $f(\overline{p}) = n\|\lambda\|_2^2/s$. Note that this implies that we can restrict to the feasbile set

$$\frac{s\lambda_i^2}{n\|\lambda\|_2^2} \leq p_i \leq 1.$$

This defines a compact region $C$. Since $f$ is continuous on this set, its maximum value is obtained at some $p^*$. $\qquad\square$

The KKT conditions then imply that at any point $p$ solving (13), we must have

$$0 \leq 1 - p_i \perp \mu - \frac{\lambda_i^2}{p_i} \geq 0, \quad i = 1, 2, \ldots, n \tag{14}$$

for some $\mu \in \mathbb{R}$. Since $|\lambda_i| > 0$ for all $i$, we actually must have $\mu > 0$. We therefore have two conditions for all $i$.

1. $p_i = 1 \implies \mu \geq \lambda_i^2$.
2. $p_i < 1 \implies p_i = |\lambda_i|/\sqrt{\mu}$.

Note that in either case, to have $p_1$ feasible we must have $\mu \geq \lambda_1^2$. Combining this with the fact that we can always select $p_1 \geq p_2 \geq \ldots \geq p_n$, we obtain the following partial characterization of the solution to (13). For some $n_s \in [n]$, we have $p_1, \ldots, p_{n_s} = 1$ while $p_i = |\lambda_i|/\sqrt{\mu} \in (0,1)$ for $i = n_s + 1, \ldots, n$. Combining this with the constraint that $\sum_{i=1}^n p_i = s$, we have

$$s = \sum_{i=1}^n p_i = n_s + \sum_{i=n_s+1}^n p_i = n_s + \sum_{i=n_s+1}^n \frac{|\lambda_i|}{\sqrt{\mu}}. \tag{15}$$

Rearranging, we obtain

$$\mu = \frac{\left(\sum_{i=n_s+1}^n |\lambda_i|\right)^2}{(s - n_s)^2} \tag{16}$$

which then implies that

$$p_i = 1, \quad i = 1, \ldots, n_s, \quad p_i = \frac{|\lambda_i|(s - n_s)}{\sum_{j=n_s+1}^n |\lambda_j|}, \quad i = n_s + 1, \ldots, n. \tag{17}$$

Thus, we need to select $n_s$ such that the $p_i$ in (17) are bounded above by 1. Let $n_s^*$ denote the first element of $[n]$ for which this holds. Then the condition that $p_i \leq 1$ for $i = n_s^* + 1, \ldots, n$ is exactly the condition that $[\lambda_{n_s^*+1}, \ldots, \lambda_n]$ is $(s - n_s)$-balanced. In particular, Lemma 2 implies that, fixing $p_i = 1$ for $i = 1, \ldots, n_s^*$, the optimal way to assign the remaining $p_i$ is by

$$p_i = \frac{|\lambda_i|(s - n_s^*)}{\sum_{j=n_s^*+1}^n |\lambda_j|}.$$

This agrees with (17) for $n_s = n_s^*$. In particular, the minimal value of $f$ occurs at the first value of $n_s$ such that the $p_i$ in (17) are bounded above by 1.

Algorithm 1 scans through the sorted $\lambda_i$ and finds the first value of $n_s$ for which the probabilities in (17) are in $[0,1]$, and therefore finds the optimal $p$ for (13). The runtime is dominated by the $O(n \log n)$ sorting cost. It is worth noting that we could perform the algorithm in $O(sn)$ time as well. Instead of sorting and then iterating through the $\lambda_i$ in order, at each step we could simply select the next largest $|\lambda_i|$ not yet seen and perform an analogous test and update as in the above algorithm. Since we would have to do the selection step at most $s$ times, this leads to an $O(sn)$ complexity algorithm.

# C Equivalence of norms

We are often interested in comparing norms on vectors spaces. This naturally leads to the following definition.

**Definition 2.** *Let $V$ be a vector space over $\mathbb{R}$ or $\mathbb{C}$. Two norms $\|\cdot\|_a, \|\cdot\|_b$ are equivalent if there are positive constants $C_1, C_2$ such that*

$$C_1\|x\|_a \leq \|x\|_b \leq C_2\|x\|_a$$

*for all $x \in V$.*

As it turns out, norms on finite-dimensional vector spaces are always equivalent.

**Theorem 10.** *Let $V$ be a finite-dimensional vector space over $\mathbb{R}$ or $\mathbb{C}$. Then all norms are equivalent.*

In order to compare norms, we often wish to determine the tightest constants which give equivalence between them. In Section 5, we are particularly interested in comparing the $\|X\|_*$ and $\|X\|_{1,1}$ on the space of $n \times m$ matrices. We have the following lemma.

**Lemma 11.** *For all $n \times m$ real matrices,*

$$\frac{1}{\sqrt{nm}}\|X\|_{1,1} \leq \|X\|_* \leq \|X\|_{1,1}.$$

*Proof.* Suppose that $X$ has the singular value decomposition

$$X = \sum_{i=1}^{r} \sigma_i u_i v_i^T.$$

We will first show the left inequality. First, note that for any $n \times m$ matrix $A$, $\|A\|_{1,1} \leq \sqrt{nm}\|A\|_F$. This follows directly from the fact that for a $n$-dimensional vector $v$, $\|v\|_1 \leq \sqrt{n}\|v\|_2$. We will also use the fact that for any vectors $u \in \mathbb{R}^n, v \in \mathbb{R}^m$, $\|uv^T\|_F = \|u\|_2\|v\|_2$. We then have

$$\|X\|_{1,1} = \left\| \sum_{i=1}^{r} \sigma_i u_i v_i^T \right\|_{1,1}$$

$$\leq \sum_{i=1}^{r} \sigma_i \|u_i v_i^T\|_{1,1}$$

$$= \sum_{i=1}^{r} \sigma_i \sqrt{nm}\|u_i v_i^T\|_F$$

$$= \sum_{i=1}^{r} \sigma_i \sqrt{nm}\|u_i\|_2\|v_i\|_2$$

$$= \|X\|_*.$$

For the right inequality, note that we have

$$X = \sum_{i,j} X_{i,j} e_i e_j^T$$

where $e_i \in \mathbb{R}^n$ is the $i$-th standard basis vector, while $e_j \in \mathbb{R}^m$ is the $j$-th standard basis vector. We then have

$$\|X\|_* \leq \sum_{i,j} |X_{i,j}| \|e_i e_j^T\|_* = \sum_{i,j} |X_{i,j}| = \|X\|_{1,1}.$$

$\square$

In fact, these are the best constants possible. To see this, first consider the matrix $X$ with a 1 in the upper-left entry and 0 elsewhere. Clearly, $\|X\|_* = \|X\|_{1,1} = 1$, so the right-hand inequality is tight. For the left-hand inequality, consider the all-ones matrix $X$. This has one singular value, $\sqrt{nm}$, so $\|X\|_* = \sqrt{nm}$. On the other hand, $\|X\|_{1,1} = nm$. Therefore, $\|X\|_{1,1} = \sqrt{nm}\|X\|_*$ in this case.

# D Hyperparameter optimization

We firstly provide results of step size tunning, as shows in Table 5 we reported stepsize tunning results for all of our experiments. We tuned these step sizes by evaluating many logarithmically spaced step sizes (e.g., $2^{-10}, \ldots, 2^0$) and evaluated on validation loss.

This step sizes tuning, for 8 gradient coding methods and 3 datasets was only possible because fairly small networks were used.

Table 5: Tuned stepsizes for experiments

| Experiments | CIFAR-10 & ResNet-18 | SVHN & ResNet-18 | CIFAR-10 & VGG-11-BN |
|---|---|---|---|
| SVD rank 1 | 0.0625 | 0.1 | 0.125 |
| SVD rank 2 | 0.0625 | 0.125 | 0.125 |
| SVD rank 3 | 0.125 | 0.125 | 0.0625 |
| SVD rank 4 | 0.0625 | 0.125 | 0.15 |
| QSGD 1bit | 0.0078125 | 0.0078125 | 0.0009765625 |
| QSGD 2bit | 0.0078125 | 0.0078125 | 0.0009765625 |
| QSGD 4bit | 0.125 | 0.046875 | 0.015625 |
| QSGD 8bit | 0.125 | 0.125 | 0.0625 |

# E Additional Experiments

**Runtime analysis:** We empirically study runtime costs of spectral-ATOMO with sparsity budget set at 1, 2, 3, 6 and made comparisons among $b$-bit QSGD and TernGrad. We deployed distributed training on ResNet-18 with batch size $B = 256$ on the CIFAR-10 dataset run with m5.2xlarge instances. As shown in Figure 4, there is a trade-off between the amount of communication per iteration and the running time for both singular value sparsification and QSGD. In some scenarios, spectral-ATOMO attains a higher compression ratio than QSGD and TernGrad. For example, singular value sparsification with sparsity budget 1 may communicate smaller messages than $\{2, 4\}$-bit QSGD and Terngrad.

Figure 4: Runtime analysis of different sparsification methods (singular value sparsification, QSGD, and Tern-Grad) for ResNet-18 trained on CIFAR-10. The values shown are computation, encoding and communication time as well as the size of the message required to send gradients between workers.

(a) CIFAR-10, ResNet-18, Best of QSGD and SVD
(b) SVHN, ResNet-18, Best of QSGD and SVD
(c) CIFAR-10, VGG11, Best of QSGD and SVD

Figure 5: Convergence rates with respect to number of iterations on: (a) CIFAR-10 on ResNet-18 of best performances from QSGD and SVD (b) SVHN on ResNet-18 of best performances from QSGD and SVD, (c) CIFAR-10 on VGG-11-BN best of performances from QSGD and SVD

Table 6: Speedups of spectral-ATOMO with sparsity budget $s$, $b$-bit QSGD, and TernGrad using VGG11 on CIFAR-10 over vanilla SGD.