[Reviews · NeurIPS 2018]

Reviewer 1



The paper is very well written. 1) it explains well the challenge of communication 2) it provides a simple yet powerful combined variance/communication objective 3) Shows how to minimize this objective, with connection to two SOTA methods for reducing the communication requirements of NN distributed training 4) Proposes a new method based on SVD that improves communication by performing a smart rotation of the axis on which the gradients are sparsified The experiments are convincing, although it would have been god to have the performance on a few other datasets. I have several slight concerns/questions: - Would it make sense to use more advanced matrix factorization technique (eg dictionary learning) to compress the gradients even more than using SVD ? The problem I foresee is the complexity of running a matrix factorization algorithm at every time step but in theory it should perform better. - The connection between the average sparsity s and the bucket size b is not clear. How are they comparable ? Minor comments -------------- It may be clearer to explicitly state a call to a sort function in Algorithm 1 Lemma 4: \exists should be in english l203: typo "we s and suppose". Why do you suppose that X is s-balanced entry wise ? For simplification ? l214: in to -> into

Reviewer 2



After rebutal; I do not wish to change my evaluation. Regarding convergence, I think that this should be clarified in the paper, to at least ensure that this is not producting divergent sequences under resaonable assumptions. As for the variance, the author control the variance of a certain variable \hat{g} given g but they should control the variance of \hat{g} without conditioning to invoke general convergence results. This is very minor but should be mentioned. The authors consider the problem of empirical risk minimization using a distributed stochastic gradient descent algorithm. In this setting, communication cost constitutes a significant bottleneck. The authors generalize recently proposed scheme which aims at obtaining random sparse unbiased estimates of a vector while ensuring minimal variance. The authors describe an active set algorithm which allows to perform this operation given decomposition in an orthonormal basis. Extensive experiments based on low rank approximation of the gradient matrix suggest that the proposed approach allows to speedup convergence by reducing the communication cost. Main comments I have very limited knowledge about distributed optimization, so I might miss some important points or shortcomings in the paper. Overall the paper is well written and seems to reference correctly the corresponding litterature. The proposed algorithm is quite natural. I did not have a careful look at all proof details but I could not see anything which looked abnormal. I am not really able to criticize numerical experiments beyond the fact that they are convincing to me. The overall mathematical exposition is quite elementary, some elements could be cut out, for example all the Lemmas related to algorithm 1 may not be necessary in the main text, this is actually a simple active set method. Another example is the notion of equivalence of norms and Theorem 10, this can be found in a first year analysis course and does not bring much to the paper. Lemma 8 is also well known. The question of the convergence of this process is not touched. How does the sparsification affect convergence? Why not comparing to ternGrad? Minor comments I am not quite sure about the relevance of the concept of "atomic decomposition". From definition 1, this is just decomposition in an orthonormal basis, which is a less fancy name but more standard as a mathematical concept. In Theorem 5, what does $j = argmax$ mean in the second statement? Similarly what is "i" in the first statement. Logical quantifiers should be used here. Figure 1, I would rather say top 2 values standing out The constraint "can be represented with k bits" is not really meaningful. Any character string can be compressed to a single bit using the appropriate dictionary. May be the authors have in mind a specific representation or model. "In what follows, you may think of g as the stochastic gradient" is not at the usual level of formality in scientific papers. Line 203: we s and suppose

Reviewer 3



This paper presents a new approach to sparsify gradients in order to reduce communication costs in distributed stochastic gradient methods. The proposed approach, called ATOMO, generalizes existing approaches of entry-wise sparsification to sparsification with respect to an atomic decomposition, where the atoms are orthonormal. ATOMO provides an unbiased estimator the gradient, which satisfy a given sparsity budget, in the chosen atomic decomposition, and which minimises variance. This generalisation is quite straightforward. Nevertheless, the authors make a worthwhile observation based on it: The authors argue that choosing an atomic decomposition other than the standard coordinate-wise one can lead to better performance. In particular, they show on some experiments that choosing SVD decomposition for sparsifying gradients in neural networks training can lead to faster training compared to QSGD; an approach which combines quantisation and entry-wise sparsification. It would be interesting to also include comparison with the other related approach of TernGrad, as well as ATOMO itself but with coordinate-wise decomposition. The paper is well-written and clear. Minor comments & typos: Eq. after line 87: missing gamma^2 before E[ ||ghat ||_2^2] Section 3: what if lbd_i = 0, are you considering 0/0 = 0, it is good to add a small remark about this. line 154: v should be g Update: As stated 1-bit QSGD is equivalent to ATOMO but only for certain sparsity levels. I think including the comparison with ATOMO with coordinate-wise decomposition is a more informative comparison between coordinate-wise and SVD sparsifications. I also agree with Reviewer 2 that some elements in the text can be cut out, such as theorem 10, lemma 8, and especially the lemmas related to algorithm which only overcomplicates a very simple method.